# Diabetes Mellitus and Pregnancy: An Insight into the Effects on the Epigenome

**DOI:** 10.3390/biomedicines12020351

**Published:** 2024-02-02

**Authors:** Andrea Meza-León, Araceli Montoya-Estrada, Enrique Reyes-Muñoz, José Romo-Yáñez

**Affiliations:** Coordinación de Endocrinología Ginecológica y Perinatal, Instituto Nacional de Perinatología, Montes Urales 800, Lomas Virreyes, Mexico City 11000, Mexico

**Keywords:** pregestational diabetes, gestational diabetes, pregnancy, epigenome, epigenetic changes, fetal programming, intrauterine development

## Abstract

Worldwide, diabetes mellitus represents a growing health problem. If it occurs during pregnancy, it can increase the risk of various abnormalities in early and advanced life stages of exposed individuals due to fetal programming occurring in utero. Studies have determined that maternal conditions interfere with the genotypes and phenotypes of offspring. Researchers are now uncovering the mechanisms by which epigenetic alterations caused by diabetes affect the expression of genes and, therefore, the development of various diseases. Among the numerous possible epigenetic changes in this regard, the most studied to date are DNA methylation and hydroxymethylation, as well as histone acetylation and methylation. This review article addresses critical findings in epigenetic studies involving diabetes mellitus, including variations reported in the expression of specific genes and their transgenerational effects.

## 1. Introduction

Diabetes mellitus is a common disease, with a steadily increasing prevalence globally. Among the factors that trigger it are obesity, poor nutrition, and a sedentary lifestyle. These risk factors have consequences in the early stages of life and also have the potential to negatively impact the adult lives of individuals. In the case of women, they have been observed to influence their pregnancies [1].

Diabetes is a frequent medical complication of pregnancy; it can be categorized as gestational diabetes mellitus (GDM) or pregestational diabetes mellitus (PGDM). Several authors have highlighted a significant increase in diabetes during pregnancy in recent decades; about 14% of pregnant women are now diagnosed as having diabetes worldwide [2].

GDM is diagnosed in the second or third trimester of pregnancy where diabetes was not clearly diagnosable before gestation. It can evolve into type 2 diabetes mellitus (T2DM) in the mother after pregnancy, especially when maternal obesity is present, although it usually disappears after childbirth [1]. In most pregnancies in which it occurs, it appears to be caused by a pancreatic response due to an inability to compensate for insulin resistance in the gestational stage [3].

PGDM status encompasses all women who have had diabetes since before conceiving (with or without a diagnosis) and is explained by the metabolic changes that occur during pregnancy due to placental lactogen, which is a hormone that carries out metabolic functions during pregnancy [4]. T2DM and type 1 diabetes mellitus (T1DM) can be distinguished as follows: T1DM is characterized by the immune system destroying pancreatic beta cells indefinitely; consequently, insulin production is very low or null. In T2DM, insulin is not metabolized correctly due to resistance to it; although the beta cells produce additional insulin, over time, the pancreas cannot produce enough insulin to maintain normal glucose levels in the body [5].

The risk of congenital diseases may increase when there is a deconcentration of maternal glucose in the first gestational weeks, as in the case of diabetic embryopathy syndrome [6], in which diabetes can have a negative effect on the fetus as discussed below. According to several authors, pregnancy has a diabetogenic effect due to increased insulin resistance, which contributes to the placental secretion of hormones such as progesterone, free cortisol, placental lactogen, prolactin, and growth hormone, among others, which provide necessary glucose to the fetus through the placenta [7]. In the case of PGDM, the deficiency in insulin production is more significant, and the intrauterine diabetic environment begins to exert its influence from the embryonic stage, producing severe periconceptional effects [8].

In recent decades, the influence of maternal conditions and the intrauterine environment on the risk of individuals experiencing certain conditions at birth and throughout life has been studied [9]. Some specialists have focused on how molecular factors alter metabolic pathways in the prenatal stage and trigger an increased risk of different diseases. Advances in genome-wide association studies [10] have allowed scientists to identify mechanisms that could be the origin of these conditions, such as epigenetic-related processes.

Epigenetics refers to all modifications that occur independently of the DNA nucleotide sequence and can influence gene expression. Environmental agents can influence epigenetic changes, so epigenetics represents a possible factor in the development of various diseases. Epigenetic marks include the methylation and hydroxymethylation of DNA, chromatin compaction by various modifications of histones (e.g., trimethylation, acetylation, and deacetylation), and the expression of miRNAs. It has been proposed that if such alterations occur during specific critical periods of early development, the epigenetic modifications generated can increase the predisposition to different pathophysiological disorders, such as metabolic syndrome, cardiovascular disease, or obesity [11].

The set of all epigenetic modifications is called the epigenome. Knowing how epigenomic changes occur and their effects is a valuable tool for studying certain diseases from a perspective parallel to genetics [12], since epigenetic changes can be reversible. In the case of diabetic embryopathy syndrome, understanding how maternal conditions impact the intrauterine environment and offspring could contribute to establishing the possible origin of diabetes during pregnancy. In addition, the early and effective identification of these epigenetic marks would facilitate taking preventive actions to reduce the risk of various diseases in individuals.

The study of epigenetic changes that occur during the prenatal stage has been of great interest in the analysis of intrauterine programming [9,13]. In this line of research, studies in humans and experimentation with animal and cell models have been essential to understanding the relationship between maternal diabetes and epigenetic alterations in progeny. The aim of this review is to provide an overview of the studies conducted relating to diabetes in pregnancy and the epigenetic changes it produces, as well as variations in the expression of specific genes and their possible short- and long-term consequences on exposed offspring. Whereas the reviewed publications could provide the basis for future research, this review is intended to contribute to the knowledge of epigenetic mechanisms and their relationship with diabetic pregnancy, as well as the potential development of therapies using the reversibility characteristic of epigenetic changes as a starting point to help prevent the deleterious effects of diabetes in the gestational stage.

## 2. Methods

Searches in PubMed and Google Scholar were performed to find relevant papers related to our objectives. The following keywords were employed: “gestational diabetes mellitus”, “pregestational diabetes mellitus”, “pregnancy”, “epigenome”, “epigenetic changes”, “fetal programming”, and “intrauterine development.” The first search was conducted in September 2021.

Articles considered for inclusion in this review had to meet the following criteria: a focus on the effects of diabetic pregnancies and other maternal conditions related to the epigenome, publications in peer-reviewed scientific journals, original studies, systematic reviews, meta-analyses, and clinical guidelines, with priority given to publications within the past five years to ensure contemporary relevance. The selection of articles was conducted in two stages. In the first stage, one of the researchers (M.L.A.) reviewed titles and abstracts, which were blinded regarding authorship, authors’ affiliations, and study results, to determine their potential relevance. In the second stage, preselected articles were read in full to ensure compliance with the inclusion criteria and to extract relevant information.

Data extraction was executed using a standardized form that included study characteristics, information on maternal conditions during gestation, epigenetic changes, and alterations in the adequate expression of specific genes. The results obtained from the selected articles were analyzed and organized for presentation in the present article. The most relevant findings were highlighted, and associations were established between maternal conditions during pregnancy and the effects caused in the epigenome of offspring. Figure 1 depicts a flow chart of the methodology applied in this review.

## 3. Results

### 3.1. Epigenetics as the Possible Origin of Diabetes during Gestation: Experimental Models and Studies in Humans

Several studies have shown that maternal nutrition is strongly related to intrauterine development, as it impacts predisposition to metabolic diseases such as T2DM [14]. Various experiments have confirmed that the excess or deficiency of nutrients in the maternal diet affects the epigenetic characteristics that occur in utero. Researchers have found that the relationship between environmental factors and their impact on epigenetics significantly affects genes linked to the development of T2DM, those responsible for the appropriate activity of beta cells and insulin resistance, and it has been proposed that such alterations lead to chromatin remodeling to its inactive state on pancreatic islets. It has also been highlighted that the relationship between environmental factors, diet, and various conditions, such as hyperglycemia or the correct metabolism of specific molecules (e.g., carbohydrates), contributes to the evolution of T2DM in the long term [15]. Early exposure to hyperglycemia increases the predisposition to develop complications linked to diabetes (referred to as metabolic memory in the literature) by developing an altered gene expression due to epigenetic modifications [16], such as DNA methylation or hydroxymethylation and changes in the chromatin. 

Notably, epigenetic mechanisms play an essential role in cell differentiation and the conservation of differentiated states, which are fundamental in the evolution of T2DM [17]. According to the same research, several epigenetic modifications occur during germ cell specialization that reprogram the DNA and histones, allowing these marks to be transmitted to offspring.

Several researchers agree that improving the study and understanding of the mechanisms entailed in epigenetic changes and their relationship with environmental variants could lead to the development of therapies to prevent the evolution of various diseases, including T2DM.

#### 3.1.1. Experimental Models

Through studying different experimental models, attempts have been made to elucidate the effect of diabetes on pregnancy; for this purpose, it has been possible to obtain animal fetuses with delayed development, malformations, and resorptions. So equipped, scientists have attempted to explain the mechanisms responsible for these alterations. Although the results obtained are diverse, it has been possible, in animal studies, to infer the mechanisms of diabetes [18].

By inducing diabetes in rats before conception, it was determined that the offspring were subsequently born with diminished fetal, craniofacial, and placental dimensions, in addition to being generally smaller than the offspring of control mothers (i.e., without diabetes) [19]. Another study concluded that the intrauterine environment in mothers with diabetes has a significant influence on offspring health by studying pregnant diabetic mice on a high-fat diet (HFD). Their findings highlighted an increased risk of malformations and death in the progeny, as well as restricted growth and alterations in metabolism in adulthood in the presence of previous hyperglycemia and during pregnancy [20]. This was supported by the fact that PGDM was found to cause certain defects even before birth in the progeny; one study reported that diabetic pregnant rats showed more fetal malformations on the 21st day of gestation, concluding that maternal metabolic changes can cause alterations during the later stages of pregnancy [21]. 

Moreover, maternal metabolic conditions affected the offspring’s development through DNA methylation or histone trimethylation, causing changes in gene expression in variants such as body weight, where mice were found to be 26.5% lighter when PGDM was diagnosed [22]. The association of an increased risk of offspring developing T2DM when mothers have GDM has been suggested based on the observation of a dysregulated expression of insulin-like growth factor 2 (Igf2) and imprinted maternally expressed transcript (H19) genes in the pancreatic islets in the offspring of mice with GDM. This dysregulation is attributed to differentially methylated regions (DMRs), which is explained by altered methylation patterns in the aforementioned genes [23]. The effect of intrauterine growth retardation (IUGR) was also studied in pregnant rats with reduced pancreatic and duodenal homeobox 1 gene (Pdx1) expression; the expression of this gene remained reduced in pancreatic beta cells, in addition to causing changes in the epigenome during embryonic development. Specific epigenetic changes included the fetal stage loss of the upstream transcription factor 1 (Usf1) binding domain in the proximal promoter, the recruitment of histone deacetylase 1, and the deacetylation of histones H3 and H4; after birth, H3K4 was demethylated and H3K9 methylated [24].

A specific diet may be associated with a long-term phenotype, and various researchers have established animal models with HFDs during the gestational stage to study their effect on the epigenome. One study highlighted that food intake was lower in diabetic progeny but that they had more visceral fat weight compared to controls. The experimental group also exhibited glucose intolerance, impaired pancreatic beta-cell function, lower interleukin concentrations, and lower insulin levels, suggesting an association between fetal programming and metabolic alterations [25]. Moreover, leptin upregulation was analyzed in the same experimental model, revealing an active period 12 days after the rat’s birth in which epigenetic remodeling occurred on the leptin promoter (decreased in 5 mC and increased in 5 hmC), which coincided precisely with the induction of the leptin gene (LEP) during adipogenesis. The observed epigenetic changes were associated with an increased expression of the LEP in offspring from the experimental mothers [26]. Using mice to study the gene expression of dopamine and opioid receptors, the authors reported lower DNA methylation in the brains of the offspring group exposed to an HFD, as well as long-term behavior that showed a predilection for nutrition-poor food, demonstrating that this type of diet is capable of producing phenotypic and epigenetic changes, such as brain decremental DNA methylation in solute carrier family 6 (neurotransmitter transporter, dopamine), member 3 (Slc6a3), opioid receptor mu 1 (Oprm1), and proenkephalin (Penk) genes [27]. In a study of the long-term impact of maternal overnutrition on epigenetic changes in the proopiomelanocortin gene (Pomc) of rat progeny, hypermethylation in the latter gene was found in individuals from the experimental group, specifically in the enhancer and promoter regions; however, only the alterations in the promoter persisted into adulthood [28]. Another investigation addressed the pancreatic characteristics of the offspring of PGDM mothers on a restricted diet of olive oil during pregnancy; five months after birth, the male offspring of diabetic mothers had reduced β-cell mass. Moreover, peroxisome proliferator-activated receptor (Pparβ/δ) and nuclear receptor peroxisome proliferator-activated receptor γ (Pparγ) mRNA exhibited decreased expression in the case of diabetic mothers’ progeny [29].

The importance of carbohydrate metabolism in the risk of diabetes through the programming of metabolic pathways has been suggested [30], as has the joint analysis of epigenetic alterations, their specific mechanisms, and genes to finally explain the phenotypic changes they cause, especially before conception. Studies have recognized that carbohydrate intake is part of the programming of mouse offspring. Controls fed with this biomolecule were reported to show a greater expression of fat mass and the obesity-associated Fto allele [31], whereas the intake of a low-carb diet during the prenatal stage prevented fetal abnormalities [32].

Trace elements and vitamins are crucial in the maternal diet during diabetic pregnancy. Using Wistar rats that were pre-conceptionally induced with diabetes by using streptozotocin, researchers observed that the offspring of untreated diabetic mothers displayed alterations in the metabolic state, redox, and trace elements, which could be linked to various adverse effects on the offspring (decreased weight, heart malformations, etc.). In contrast, diabetic mothers fed with elements such as zinc significantly improved [33], leading to the conclusion that insufficient pregestational and gestational zinc intake can influence in utero programming in offspring [34]. Additionally, pregnant sheep with vitamin B and methionine restriction produced offspring with characteristics such as elevated blood pressure, abnormal immune response, and insulin resistance (most notably in males). After a scan of the fetal liver genome using restriction landmark genome scanning, an autoradiograph was obtained showing spots representing unmethylated CpG sites (which were digested with NotI, a methylation-sensitive restriction enzyme), thus providing information on the phenotype of the study model [35].

#### 3.1.2. Studies in Humans

Several studies in humans have hypothesized that changes in the epigenome that occur during fetal development because of variations in the uterus may be among the determinants of chronic diseases in adulthood, including T2DM; moreover, intrauterine development conditions may be related to an increased or decreased predisposition to diverse chronic diseases [36,37]. 

Maternal glycemia levels during early pregnancy have been associated with altered fetal growth and disturbed glucose metabolism in childhood. In mid-pregnancy, higher glucose levels are associated with decreased fetal growth, whereas in late pregnancy, they are linked to increased fetal growth [38], exacerbating the risk of cardiometabolic conditions in offspring when higher maternal insulin levels are present during early pregnancy. Furthermore, an increased risk of childhood overweight related to higher maternal levels of insulin and glucose at the same stage has been found [39]. 

Concerning diabetes in pregnancy and neurodevelopmental disorders in children, beyond the fact that children present perinatal complications, the prevalence of learning disorders, attention deficit, hyperactivity, and autism spectrum disorders increases in the long term [40]. In addition, an investigation revealed three significant results concerning the origin of T2DM by studying gestational impaired glucose tolerance (IGT): (1) low DNA methylation on the C1Q adiponectin and collagen domain containing (ADIPOQ) promoter on the fetal side of the placenta is associated with high levels of maternal glucose during the second trimester of pregnancy; (2) low DNA methylation on the maternal side of the placenta is associated with a higher rate of insulin resistance during the second and third trimesters; and (3) low DNA methylation levels are associated with higher levels of adiponectin during pregnancy. Since adiponectin is thought to have characteristics that induce glucose sensitivity, these epigenetic variations could lead to changes in glucose metabolism in the mother and child [41]. By studying placental and maternal tissue and umbilical cord blood samples, researchers determined that DNA methylation levels in the LEP of the placenta are linked to glucose levels in women with IGT and a decreased expression of the LEP, supporting the conclusion that IGT is associated with DNA methylation and that together they have a potential functional impact [42], thus establishing precedents that would explain conditions associated with fetal programming.

Furthermore, the impact of maternal diet on children has been observed. A low-nutrient diet and low maternal weight increase the risk of developing GDM and, therefore, predispose children from such pregnancies to developing T2DM [43].

A study of progeny through six years of life confirmed that the alteration of vitamin B intake during pregnancy led to long-term effects in offspring of the studied women [44]. Another study analyzed a group of individuals periconceptionally exposed to the 1944 Dutch famine and observed that the IGF2 gene still exhibited lower methylation compared to controls six decades after the famine, thus demonstrating that the early stages of development have an essential function in the presence of epigenetic marks and that those marks can persist well into the individual’s adult stage [45]. Another study yielded similar results; however, the effects were more significant in male individuals [46]. Both studies could be taken as points of departure to reaffirm the proposition that epigenetic changes that occur during pregnancy may be the possible cause of alterations in mature stages of life. Moreover, the connection between the maternal diet and the offspring’s body composition is related to epigenetic changes in the genes responsible for metabolic control by increasing methylation, which is associated with a low carbohydrate intake in early pregnancy. Perinatal epigenetic analyses can be used to identify the vulnerability of individuals to certain metabolic diseases, such as T2DM [47]; this highlights the importance of early diagnosis through the use of early genetic trials that help establish a relationship between genome variations and individual phenotypes [48]. 

Several researchers have studied how fetal exposure to GDM affects gene expression, as in a recent study in which variations in gene expression were identified by observing DNA methylation changes in arterial endothelial cells and venous endothelial cells as well as alterations in actin organization and barrier function, which indicate a clear association with the development of diabetes [49]. Likewise, in a study looking for DMRs, two hypomethylated regions were found in newborns exposed to GDM: one includes the promoter of olfactory receptor family 2 subfamily L member 13 (OR2L13) (a gene associated with autistic spectrum disorder), and the other is in the gene body of the cytochrome P450 family 2 subfamily E member 1 (CYP2E1) gene, which is linked to the regulation of T1DM and T2DM, thus establishing a relationship between DMRs and the abovementioned maternal condition [50]. Further supporting this association, even minor changes in umbilical cord blood methylation can be attributed to several genes and loci, which are possible transmitter candidates of the effects of GDM on progeny [51]. It has also been reported that children of mothers with this disease are more likely to have cardiometabolic conditions, in addition to a high level of insulin in the umbilical cord (intrauterine hyperinsulinemia), which is suggested as a determinant for predicting tolerance to abnormal glucose during childhood and could be a factor in the development of T2DM [52]. In addition, higher incidences of heart and valve defects have been observed in exposed children, suggesting that these are genetically associated with the risk of cardiovascular and cerebrovascular conditions [53].

Researchers that aimed to identify the epigenetic metabolic pathways affected by GDM by using samples of placenta and umbilical cord blood found that gene methylation was potentially different in exposed individuals compared to controls. This difference was also associated with the weight of newborns, suggesting its influence on fetal growth and development. This was also presented as evidence that DNA methylation has a relationship with fetal metabolic programming [54]. In addition, 98 DMRs associated with this maternal condition have been identified, suggesting that epigenetic marks can indicate a link between the intrauterine environment and early obesity in exposed offspring [55], as well as T2DM and long-term cardiovascular diseases [56,57]. Research has also shown that epigenetic changes influence metabolic programming during in utero development, explicitly finding different methylation in the DNA at the lipoprotein lipase (LPL) locus compared to that in control patients [58].

Another study compared the abdominal subcutaneous and omental visceral adipose tissues of women with GDM and those of control women. No changes in methylation in the putative peroxisome proliferator response element 2 (PPRE2) in mothers with GDM were noted, which supports the proposition that DNA methylation does not have a direct regulatory outcome, thus suggesting that chromatin modification (mediated by histones) could be responsible for the effect on transcription activation or deactivation in a tissue-specific manner [59].

T2DM and T1DM also cause epigenetic changes during pregnancy. In a cohort studied up to age 35 years that was exposed to GDM and T2DM, primarily significant morbidity related to cardiovascular disease was noted [60] due to the in utero environment being capable of causing damage to the body composition and vascular health of the exposed offspring when maternal hyperglycemia occurred [61]. By looking for promoters with DMRs and then subjecting them to a Kyoto Encyclopedia of Genes and Genomes route analysis in peripheral blood leukocytes from two nondiabetic groups of Pima Indians, one control group and the other descended from diabetic mothers (some of whom were diagnosed with diabetes prior to the first test of the study), it was found that one of the more differentiated pathways is associated with T2DM, which supports the approach that epigenetic changes increase the risk of developing T2DM due to the functional involvement of beta cells in the children of mothers with diabetes during pregnancy, in addition to showing that certain ethnicities have a greater predisposition to T2DM [62]. Generally, certain complications, such as macrosomia, during pregnancy can be associated with the presence of PGDM [63]. It has been determined that women with uncontrolled pre-conceptional diabetes exhibit a higher incidence of children with moderate to severe alterations, among which digestive, cardiovascular, and central nervous system irregularities are prevalent [64], implying that PGDM is associated with an increased risk of stillbirth, congenital malformations, and problems attributed to premature birth [65].

The epigenetic significance of nicotinamide N-methyltransferase (NNMT) has recently been suggested in obesity and T2DM. NNMT is an enzyme involved in the one-carbon metabolism and the catalysis of nicotinamide methylation via S-adenosyl methionine for the regulation of nicotinamide adenine dinucleotide (NAD+) cellular levels [66]. It plays an essential role in energy metabolism and the development of diabetes. The inhibition of NNMT promotes weight and adipose tissue loss, in addition to the adequate metabolism of glucose and insulin. However, the metabolic pathways in which this molecule participates have not been elucidated. NNMT activity has been observed to increase in patients with T2DM, who also exhibit a higher expression of NNMT [67]. Decreased methylation was linked to the increased expression of NNMT in obese patients in the following genes: collagen type XXIII alpha 1 chain (COL23A1), plectin (PLEC1), F-box protein 21 (FBXO21), STEAP3 metalloreductase (STEAP3), regulator of G protein signaling 12 (RGS12), immunoglobulin superfamily DCC subclass member 3 (IGDCC3), forkhead box K2 (FOXK2), and ORAI calcium release-activated calcium modulator 2 (ORAI2) [68]. Furthermore, the use of inhibitors in treating metabolic disorders such as obesity and diabetes has been suggested via using small molecules such as JBSNF-000028 to inhibit NNMT activity, thereby improving glucose tolerance after treatment [69]. The deletion of NNMT improves glucose tolerance and insulin sensitivity in type 2 diabetic mice [70].

### 3.2. DNA Methylation in Diabetic Pregnancies: Cellular and Animal Models, and Studies in Humans

Changes in DNA caused by diabetes during the prenatal stage play an essential role in the appropriate regulation of the transcriptional process and gene expression in the offspring, which is known to be related to the risk and subsequent development of various diseases and conditions, such as insulin resistance, diabetes, and heart disease, among others. Several authors have observed epigenetic alterations in the DNA of the placenta and umbilical cord in diabetic pregnancies compared to control groups; such changes could also remain with the individual in the long term [13,14,71].

#### 3.2.1. Cell Models

Maternal diabetes causes changes in the epigenome and, therefore, in the expression of genes involved in developing the neural tube. The basis of neurodevelopmental disorders in the progeny of diabetic pregnancies has been established by analyzing human neural progenitor cells (hNPCs) exposed to high glucose concentrations in a medium. The search for alterations in genes due to changes in DNA methylation has uncovered that they can cause alterations in neural tube formation, which impacts the neurodevelopment of exposed individuals. Those hNPCs exposed to high glucose concentrations show changes in DNA methylation in specific genes associated with biological pathways, such as the SLIT1-ROBO2 pathway (which mediates neurogenesis and cell proliferation), and in Hippo pathway genes, such as Yes-associated protein (YAP) and WWTR1 (TAZ), which are involved in proliferation, stemness, differentiation, and organ size. Hypomethylated regions were found in the 5′UTR and the body of the YAP gene, as was increased and decreased methylation in the YAP gene. The expression of the abovementioned genes was downregulated [72]. 

DNA methylation assays were conducted in search of an effect caused by GDM on gestational biological age in children at birth through comparison with the Knight gestational epigenetic clock, a test used to estimate gestational age based on DNA methylation. GDM was reported to have potentially deleterious effects on the health of newborns, causing macrosomia, hypoglycemia, and respiratory distress syndrome, among other conditions. Umbilical cord blood cells from exposed newborns were collected for the study, specifically granulocytes, monocytes, natural killer cells, B cells, nucleated red blood cells, CD4 cells, and CD8 cells. The DNA methylation level was related to reduced biological maturity at birth versus the offspring of control pregnancies [73]. Another study addressed DNA methylation in the pancreatic islets of human donors under hyperglycemic and control conditions to examine the effects of high levels of glucose, which have been observed as having the potential to change the expression of genes associated with glucose metabolism and to affect insulin secretion compared to healthy individuals. In the cultures with high concentrations of glucose, DNA methylation showed a more significant increment compared to control cultures; additionally, the glycine receptor alpha 1 (GLRA1) and VAC14 component of the PIKFYVE complex (VAC14) genes showed decreased expression, while the ras-related dexamethasone-induced 1 (RASD1), solute carrier organic anion transporter family member 5A1 (SLCO5A1), and cholinergic receptor nicotinic alpha 5 subunits (CHRNA5) genes exhibited increased expression [74].

Another interesting study was conducted using streptozotocin injections, a drug that promotes the destruction of pancreatic beta cells, leading to the induction of diabetes. Pregnant mice were used to look for epigenomic changes, and only male mice were selected for further analysis of the primordial germ cells (from day 13.5) of three generations. Bisulfite sequencing showed methylated genes enriched in obesity and diabetes. Interestingly, methylation validation of the Fyn proto-oncogene (Fyn) showed hypomethylation in the F1 primordial germ cells and F2 somatic cells, indicating that epigenetic changes can occur in early pregnancy and intergenerationally (Figure 2) [75]. 

#### 3.2.2. Animal Models

Mouse and rat models are used most widely in this field. As noted above, diabetes is induced in these animals via injections of streptozotocin before or after copulation to study GDM or PGDM and their relationship with the epigenome.

In a study of the effect of GDM in in utero programming, by analyzing the offspring and, above all, the results, global DNA hypermethylation in the exposed offspring was found in rat hearts; this is significant, as increased levels of reactive oxygen species have been associated with the GDM condition and the development of cardiovascular pathologies in exposed offspring. It was concluded that cardiac oxidative stress and altered hypermethylation were generated by the maternal condition, which caused the downregulation of the sirtuin 1 (Sirt1) gene [76]. Also, while looking for alterations in the metabolic phenotype and DNA methylation in the pancreas using the same experimental model, the authors found that the intrauterine environment influenced the development of dyslipidemia, insulin resistance, and glucose intolerance. In another study, cardiac alterations were evaluated by inducing hyperglycemia in mice before copulation and contrasting them with a control group. This was motivated by the fact that this condition has been observed to change patterns in gene expression in processes such as cardiac neural crest cell migration, outflow, and inflow tract formation. A massively parallel sequencing-based methylation-sensitive restriction-based assay was performed to analyze more than 1.65 million loci on day 0 of newborns and was validated with RT-qPCR. In histological analysis, heart defects were observed in 28% of the offspring exposed to hyperglycemia, in contrast to controls (7%). Moreover, increased DNA methylation in the promoter regions of genes linked to cardiac development was observed [77]. In another study, the hearts of GDM mice were used, as the altered expression of specific genes has been related to the development of congenital heart diseases, such as ventricular septal defects. The study aimed to discover whether cardiac development is altered by DNA methylation. Abnormal methylation was reported in the group of exposed embryos versus controls in the small ubiquitin-like modifier 3 (Sumo3) and pyruvate dehydrogenase phosphatase catalytic subunit 1 (Pdp1) genes, which may also have implications for correct cardiac development as a consequence of the variation in gene expression (Figure 2) [78]. 

Numerous DMRs, which are involved in glycolipid metabolism, have been observed in the pancreas of a GDM model. A study revealed elevated methylation in the ArfGAP with GTPase domain, ankyrin repeat, and PH domain 2 (Agap2) gene, which was found to be upregulated, suggesting a connection between these changes and an increased risk of T2DM and obesity in the adulthood of the exposed offspring caused by the induction of hyperinsulinemia during the fetal stage [79]. Using the pancreatic islets of Wistar rat offspring and analyzing them with bisulfite sequencing to find patterns in the DNA methylation of CpGs in the promoter regions of cyclin-dependent kinase inhibitor 2A/B (Cdkn2A/B), the methylation level of the Cdkn2A promoter was lower in the GDM-exposed offspring, implying that this effect could alter epigenomic characteristics in the Langerhans islets by causing decreased β-cell mass proliferation and mild hyperglycemia, which could be the origin of diabetes (Figure 2) [80].

In another study, mice were fed an HFD with the aim of identifying epigenetic changes. This approach was motivated by the determination that this type of diet during pregnancy can cause conditions such as an increase in normal weight, hyperglycemia, and alterations in insulin resistance. Of all the genes affected by the maternal diet, about 25–33% had DMRs, supporting the conclusion that an HFD in the gestational stage can cause variation in DNA methylation, reduced insulin sensitivity, and altered expression in more than 3900 liver genes [81], as well as an increased risk of conditions such as diabetes and obesity as a consequence.

In addition, an elevated risk of infertility in adulthood has been reported in mice exposed to GDM due to a decrease in 5-methylcytosine (5 mC) and an increase in 5-hydroxymethylcytosine (5 hmC) levels in the cocaine- and amphetamine-regulated transcript Cart prepropeptide (Cartpt) promoter, which together cause leptin sensitivity to increase in the abovementioned promoter and an increased risk of ovarian dysfunction by affecting folliculogenesis [82]. Furthermore, rats have been used to test the effect of folic acid and vitamin E in the uterus. The trial showed that the uterus of diabetic rats had lower methylation in the transforming growth factor beta 1 (Tgfb1) promoter, which could potentially increase its gene expression. Tgfb1 has been reported to regulate the proper conditions of epithelial and endometrial tissues. Therefore, folic acid and vitamin E were proposed as a pathway to counteract the consequences of diabetes in utero (Figure 2) [83]. 

#### 3.2.3. Studies in Humans

Research in cell and animal models has led to breakthroughs in knowledge in the field of epigenetics, but numerous studies in humans have also helped to understand the importance of diabetes as one of the most common pregnancy complications, as well as its short- and long-term effects.

A study of the offspring of mothers with GDM and T1DM showed increased methylation in the resistin (RETN) gene and weaker expression in the T1DM group. In addition, GDM was associated with increased methylation in LEP and ADIPOQ and reduced expression in LEP, ADIPOQ, and RETN. Variations in LEP levels are reportedly associated with metabolic alterations in individuals exposed to GDM, and ADIPOQ levels have been associated with T2DM risk [84]. Furthermore, using a multivariate regression model, researchers have found significant differences in the methylation of certain genes, such as microfibril-associated protein 4 (MFAP4), protein kinase C eta (PRKCH), solute carrier family 17 member 4 (SLC17A4), and hypoxia-inducible factor 3 subunit alpha (HIF3A); the alterations were validated with bisulfite pyrosequencing (Figure 3). The results remained unchanged with adjustments for factors such as maternal body mass index, gestational weeks, and the fetus’s sex. The methylation effects on individuals exposed to insulin-dependent GDM were more noticeable than in those exposed to GDM whose mothers were dietetically treated [51]. 

In other research, DNA methylation in peripheral blood in offspring from mothers with GDM and control patients in the Danish National Birth Cohort aged between 9 and 16 years was measured using Illumina HumanMethylation450 BeadChip for the analysis, with pyrosequencing to validate the results. Blood evidenced the epigenetic characteristics of specific tissues, and the approach was proposed as having potential use to calculate the risk of developing different disorders in the short and long term. In GDM offspring, 76 differentially methylated CpGs were identified (compared to controls); however, 13 changes were associated with this condition. Pre-pregnancy BMI-associated CpGs (cg00992687 and cg09452568 of the endothelial cell-specific molecule-ESM1 and cg14328641 of the membrane spanning 4-domains A3-MS4A3) were validated, while cg09109411 (phosphodiesterase 6A-PDE6A) was linked with maternal GDM status (Figure 3) [85]. Also, in the analysis of DNA methylation and adiponectin alterations in mothers and children exposed to GDM, it was found that mothers exhibited hypoadiponectinemia associated with enlarged maternal insulin resistance in the perinatal stage. In the mother groups, a significant decrease in mRNA was observed in subcutaneous and visceral adipose tissue, with minor variations in DNA methylation. The offspring groups presented several methylation variations and similar adiponectin values, leading to the conclusion that reduced adiponectin may influence the development of GDM, affecting maternal–fetal metabolism [86].

In the fetal placenta side of women diagnosed with GDM and control pregnancies, it was observed that in the experimental group, the DNA methylation of the serotonin transporter gene SLC6A4 (solute carrier family 6 member 4) was increased in the controls, indicating that the in utero environment can alter the serotonin system due to this neurotransmitter being involved in growth and neurodevelopment in utero. It has also been linked to the development of obesity and metabolic disorders [87]. In a different study, cord blood from newborns was used, and an association between GDM and alterations in DNA methylation patterns was discovered; two DMRs were found in exposed newborns, both with decreased methylation compared to control subjects: one DMR in the promoter of olfactory receptor family 2 subfamily L member 13 (OR2L13), which has been found in blood and buccal cells from patients with autism spectrum disorder, and the other DMR in the gene body of cytochrome P450 2E1 (CYP2E1) [50], which is linked to T1DM and T2DM (Figure 3).

The placentas of Native American and Hispanic women with diabetes during pregnancy were compared with the placentas of healthy women to find alterations in DNA methylation. In a genome-wide DNA methylation analysis, 247 CpG sites with significant changes were found for female offspring, 465 for male offspring, and 277 for both sexes. Additionally, specific loci with changes in DNA methylation were detected and associated with mitochondrial function, DNA repair, inflammation, and oxidative stress in specific genes: piwi-like RNA-mediated gene silencing 3 (PIWIL3), cytochrome b-245 alpha chain (CYBA), glutathione S-transferase mu 1 (GSTM1), glutathione S-transferase mu 5 (GSTM5), potassium voltage-gated channel subfamily E regulatory subunit 1 (KCNE1), and nucleoredoxin (NXN) (Figure 3) [88].

DNA methylation has been proposed as a viable option for diagnosing diabetic embryopathy syndrome. A study that used bisulfite sequencing was conducted to identify DMRs in newborns with this syndrome due to the presence of diabetes in pregnancy versus control newborns; 237 loci were recognized with specific changes in the methylation of the diabetic embryopathy syndrome neonates, which are close to genes associated with Mendelian alterations linked to the diabetic embryopathy syndrome phenotype, such as calcium voltage-gated channel subunit alpha1 C (CACNA1C), ankyrin repeat domain-containing protein 11 (ANKRD11), or genes known to be capable of influencing embryonic development, such as RAS p21 protein activator 3 (RASA3) (Figure 3) [89]. 

We found no studies following up the epigenetic changes in PGDM- or GDM-exposed offspring from birth and during different stages of life, including childhood and adulthood; however, groups of children between 3 and 10 years after birth have been investigated. GDM-exposed offspring were analyzed using an epigenetic clock, and exposure to GDM was related to increased DNA methylation compared to the children of nondiabetic mothers [90]. Also, a group of 5-year-old children exposed to hyperglycemia during the gestational stage was studied, and a decrease was found in the methylation of the gene fibronectin type III and SPRY domain-containing 1 like (FSD1L) [91]. 

The association between HbA1c and DNA methylation in cord blood was investigated, and elevated levels of HbA1c during the second trimester of pregnancy were associated with lower methylation in the upregulator of cell proliferation (URGCP) gene [92]. Additionally, an adult group exposed to GDM and obesity in utero was analyzed; it was found that higher levels of HbA1c are related to an increased expression of tetraspanin 14 (TSPAN14) in subcutaneous adipose tissue [93]. Table 1 summarizes some of the findings mentioned above, along with the conditions studied in each investigation.

### 3.3. Acetylation, Methylation, and Other Epigenetic Modifications in Histones: Cell and Animal Models and Studies in Humans

Although most studies addressing the effect of diabetes on pregnancy have focused on epigenetic changes in DNA, the condition can also alter histone proteins. Histones can undergo specific changes, such as acetylation, methylation, or phosphorylation, which can promote or inhibit gene expression levels [94] and chromatin conformation (euchromatin or heterochromatin); therefore, they can modify the transcription.

#### 3.3.1. Cell Models

Cellular models have been established to study possible implications and repercussions of diabetes in the gestational stage at the histone level. The relationship between GDM and histone modifications in progeny has been studied using peripheral blood mononuclear cells, umbilical vein endothelial cells, and cord blood mononuclear cells from newborns of GDM and control pregnancies. An upregulation of the nuclear factor kappa-light-chain-enhancer of activated B cells p65 (NF-kB p65) gene was found in the diabetic cells compared to controls via trimethylation of the histone H3K4 (H3K4me3) (Figure 3). This was probably due to the inflammation and oxidative stress caused by the hyperglycemic condition during pregnancy [95]. Moreover, trophoblast cells were elected to identify histone modifications in forkhead box O1 (FOXO1) by using the choriocarcinoma cell line BeWo and primary human villous trophoblast cells, which were treated with calcitriol (the active form of vitamin D, the deficiency of which has been associated with insulin resistance in muscle cells). H3K9ac was found to be downregulated in the placentas of mothers with GDM, while the H3K9ac of cell cultures from control pregnancies was also negatively regulated in calcitriol treatment, suggesting that epigenomic alterations are involved in GDM. These are not adequately treated with calcitriol [96].

Twelve-month-old mice with multiparity-induced diabetes mellitus, which has been proposed to be associated with T2DM, were studied by integrating histone methylation analysis in beta cells from experimental and control models to determine specific genes for dedifferentiation, such as vitamin D-binding protein (Gc). In mice deficient in this gene, an increased insulin response was observed under hyperglycemic conditions compared to controls, suggesting the importance of Gc in the correct function of beta cells. Differences were also found in histone H3K4, such as an increased trimethylation of Rap guanine nucleotide exchange factor 5 (Rapgef5), ETS homologous factor (Ehf), and Gc, as well as a decreased trimethylation of hematopoietically expressed homeobox (Hhex), adenylate cyclase 5 (Adcy5), and LIM domain only 4 (Lmo4) genes in diabetic models [97]. In addition, to explain the maintenance of beta cells, a model of T2DM mice was studied because it has been observed that the expression of genes associated with insulin production is affected by metabolic stress. The trimethylation of histone H3K4 in Pdx1 and that of solute carrier family 30 (zinc transporter), member 8 (Slc30a8) were decreased, while augmentation was found in aldehyde dehydrogenase family 1, subfamily A3 (Aldh1a3), and lactate dehydrogenase A (Ldha) compared to the control group [98], which influences the proper functioning and identity of beta cells (Figure 2). 

The relationship between altered histone lysine malonylation and neural tube defects (NTDs) associated with PGDM has also been analyzed by inducing diabetes in mice using two streptozotocin intraperitoneal injections in order to compare the brain cells of control and diabetic mice. Higher histone malonylation levels were found in cultures of diabetic mice [99], supporting the existing connection between diabetes, environmental conditions, and NTDs.

#### 3.3.2. Animal Models

Studies have addressed the effect of diabetes on pregnancy in different animal models, especially murine models. It has been observed that the effects of an HFD during pregnancy can be offset by maternal physical activity, thereby improving glucose metabolism in the liver of offspring. In one study, a dysregulation of liver glucose metabolism was noticed in the offspring of mice with high fat intake together with the deactivation of H3K4 methyltransferase, which led to a reduced H3K4me3 level at the promoters of genes related to glucose metabolism, such as phosphofructokinase, liver, B-type (Pfkl), pyruvate dehydrogenase E1 alpha 1 (Pdha1), oxoglutarate (alpha-ketoglutarate) dehydrogenase (lipoamide) (Ogdh), acyl-Coenzyme A oxidase 1, palmitoyl (Acox1), and carnitine palmitoyltransferase 1a, liver (Cpt1a). The alterations were restored in the offspring of mothers who exercised during the prenatal stage [100]. 

The association between maternal diabetes and an HFD was studied, as these maternal conditions can cause lipotoxic effects in the hearts of exposed offspring, leading to the development of altered cardiac function. One study analyzed the offspring of rats under these conditions, and interestingly, differences in acetylation were found in histones H3K9 and H3K14, as well as the trimethylation of histones H3K4 and H3K27 in offspring exposed to diabetes during pregnancy and in the offspring of mothers fed an HFD in the gene heat shock protein 1A (Hspa1a) (Figure 2) [101], which indicated that the maternal condition is capable of producing epigenetic marks in histones related to susceptibility to cardiac pathologies. By using the same experimental model, neurodevelopmental disorders in the progeny hippocampus were studied because maternal obesity during the gestational stage in humans has been reported to be associated with anxiety, depression, decreased learning skills, and autism through epigenetic regulation. It was concluded that maternal nutrition during pregnancy can damage the offspring’s brain epigenome (more significantly in male individuals) through alterations such as an increment in the binding of the active histone mark H3K9ac at the transcriptional start site of the oxytocin receptor (Oxtr) in the male offspring’s hippocampus [102]. 

Folliculogenesis in the ovary was found to be altered, indicating that epigenetic changes occurred during the fetal stage, such as miR-101 induced by glucose and insulin, and the phosphatidylinositol 3-kinase/Akt signal transduction pathway, affecting Cartpt. Together, they are responsible for regulating the enhancer of zeste 2 polycomb repressive complex 2 subunit (Ezh2), which promotes H3K27me3 and CBP/p300, which in turn promotes H3K27ac, causing the Cartpt promoter to increase its sensitivity to leptin and predisposing female offspring from diabetic pregnancies to an increased risk of infertility during adulthood (Figure 2) [82]. 

The use of streptozotocin to induce diabetes in animal models has not been the exception in studies addressing epigenetic changes in histones. The effect of maternal hyperglycemia on offspring was analyzed in mice, specifically on neocortical neurogenetic differentiation, based on the fact that the exposed offspring of humans have an increased risk of neurodevelopmental disorders caused by damage to the neurogenesis process. Alterations in the histone acetylation of the experimental models subjected to a high glucose concentration were observed in the H3K14 histone. It was also found that maternal metabolic conditions can increase the expression of histone acetylase P300 and decrease the concentration of histone deacetylase sirtuin 1 (Sirt1) [103]. In embryos exposed to PGDM, NTDs and premature senescence were observed in the neuroepithelium; neurulation disruptions in offspring exposed to diabetes (versus controls) were also noted, in addition to a decrease in the signal from H3K9me3 due to the deletion of the Foxo3a gene, suggesting that epigenetic changes caused by the condition of maternal diabetes can lead to the development of NTDs [104].

Female PGDM mice were studied to analyze the biochemical changes that occur in the presence of maternal hyperglycemia. Among the results obtained, the offspring of diabetic mothers, compared to controls, showed DNA damage and, thus, the co-localization of DNA-damage histone γH2AX marker (pSer139) and O-GlcNAcylation, which supports findings indicating that the intrauterine environment is capable of generating alterations in the offspring, in addition to causing perinatal complications such as neonatal hypoglycemia [8].

#### 3.3.3. Studies in Humans

Authors have used human samples to establish a connection between maternal diabetes and changes in the epigenetic histone modifications of offspring, leading to the development of different conditions both immediately and over time.

By collecting placenta tissue from the fetal side, the association between maternal insulin resistance and epigenetic changes in the genome was addressed; based on its role in transferring nutrients to the fetus, an increase in the histone modifications in the Encyclopedia of DNA Elements histone modifications database, predominantly caused by H3K27me3, suggests that the epigenetic changes (including DNA and histone modifications) are related to maternal insulin sensitivity in the prenatal stage (Figure 3) [105]. In addition, in research using placental explants from T2DM and GDM mothers (given the susceptibility of this organ at the mitochondrial level of metabolic stress caused by diabetes, which can evolve into oxidative stress causing alterations in the offspring), the effect on mitochondrial biogenesis and metformin treatment was investigated. In the first instance, a decreased histone acetylation (H3K27 acetylation) was observed in placentas from both groups of diabetic mothers, as well as an increased promoter methylation of peroxisome proliferator-activated receptor gamma coactivator 1-alpha (PPARGC1A), compared with control subjects. The study determined that metformin can restore some epigenetic alterations caused by diabetes [106].

Peripheral blood plasma from 187 mothers with T2DM and GDM was also used to look for proteins associated with fetal malformations. The expression of the neurotrophic factor DFP3 of children from diabetic mothers was reduced compared with the control group; DFP3 is associated with alterations in the central nervous system, as well as in neurogenesis and myogenesis processes, and is also known to participate in the regulation of chromatin remodeling by binding to certain acetylated and methylated histone regions [107].

By using bioinformatics, umbilical cord gene expression profile datasets from mothers with T1DM and control patients were analyzed with the goal of establishing the effect of T1DM on gene expression, as endothelial cells of the umbilical cord show alterations in capillarity, the formation of cell colonies, and renewal capacity in diabetic pregnancies. Euchromatic histone-lysine methyltransferase 1 (EHMT1) was upregulated in samples from patients with T1DM compared with those of control patients [108], which indicates that T1DM can generate histone-level changes (Figure 3).

The regulation of histone deacetylase 2 (HDAC 2) and its relationship with mitochondrial function and cytokine secretion were also studied, given that there is damage to the regular mitochondrial activity in patients with GDM. For this purpose, peripheral blood monocytes/macrophages from women diagnosed with GDM and controls were used. In the experimental group, the expression of HDAC2 had a lower regulation level compared to that of the control group, which could be associated with cytokine secretion and improper mitochondrial function (Figure 3) [109]. Table 2 condenses some of the investigation results for histones.

## 4. Discussion

The interest of the present review was to analyze the literature on epigenetic modifications, specifically on DNA methylation and hydroxymethylation and histone methylation and acetylation, and their relationship with diabetes in pregnancy, and to find out whether these epigenetic modifications have an association with changes in the expression of specific genes in cellular and animal models and in humans. Also considered were the tissues or cell types the studies used and the stage of development investigated, whether during embryonic or fetal development, at birth, during childhood, or until adulthood. As not all the studies addressed all these criteria simultaneously, we sought to provide a broad outline, with Figure 2 and Figure 3 and Table 1 and Table 2 providing ready references.

Several of the reviewed studies arrived at the same determination: diabetes during pregnancy can change an individual’s epigenome [20,22,62]. In general, analyses in experimental models and studies in humans have yielded common findings, such as alterations observed in the phenotype and malformations in individuals exposed to diabetes in the gestational stage, as well as changes in metabolism in the short and long term [6,38,53], which is also an essential factor in the development of various conditions. Furthermore, variations in the expression of specific genes due to disturbances in the transcriptional process caused by the presence of epigenetic changes (especially genes associated with glucose metabolism), in addition to morphological anomalies in pancreatic islets, when coupled with the intake of specific molecules as part of the maternal diet during pregnancy (in minimal or excessive amounts), were associated with an increase or decrease in changes in the epigenome [9,28,81,96]. It is also important to highlight that some molecules, such as NNMT, were observed in metabolic disorders such as T2DM and obesity; however, their relationship with GDM and PGDM still needs to be explored to understand how they interfere with the adequate functioning of metabolic pathways and their potential in therapy to treat these conditions [67,69].

The study of the epigenome can be considered a relatively recent discipline compared to some of its counterparts; consequently, the field of research remains wide. Analyzing the metabolic pathways that trigger epigenetic changes could be valuable in investigating the effects of diabetes on pregnancy, especially if their relationships with different environmental factors, such as diet, lifestyle, and even pollution, are considered.

The experiments carried out to date in animals are numerous and diverse; however, the mouse remains the most studied organism in this area, and it is necessary to increase efforts in studies in humans to obtain knowledge of epigenetics beyond the established models [19,21,31]. For example, a mouse’s lifespan is considerably shorter than that of a human; the effect of environmental factors on the epigenome of the human being is surely greater. Therefore, it will be essential to use existing animal and cellular models to expand the panorama to human studies. Furthermore, implementing more robust experimental and human studies may help support the relationship between alterations in the epigenome due to PGDM or GDM.

Taking as a priority comprehensive studies of epigenetics and its relationship with diabetes, more attention could be paid to the morphology of pancreatic islets [23], the metabolic pathways of insulin, and how they are affected by the presence or absence of epigenetic changes. Additionally, much current research has focused on studying genes known to be related to diabetes; it would be intriguing to analyze how variations in the expression of these genes are interrelated with others not yet associated with this condition.

It is essential that specialists recognize the epigenome’s influence on an individual’s life [15,16]. Epigenetic changes caused by diabetes could be considered as potential molecular markers for predicting adverse effects, although researchers should also seek to replicate the results already reported, expand study models, and increase patient sample sizes to enable more robust and sustained determinations. Researchers might also start to consider paternal factors as a control to determine whether epigenetic changes in offspring are caused by the maternal condition in the gestational stage and not by a set of hereditary factors originating from both parents or from related diseases such as obesity or metabolic syndrome. Drugs that work on the epigenome are already being studied as possible therapies to reverse the effects caused by diabetes [69,70]. However, other interventions have been proposed to prevent damage caused by epigenetic changes in diabetic pregnancy, such as eating a balanced diet, weight control, physical activity, or glycemic control [16,43,96,97]. More studies are needed showing that decreases in epigenetic changes due to the maternal lifestyle (for example, diet or exercise) also cause, as a consequence, improvement in the health of the offspring in the long term [110]. Although great discoveries have been made in the field during the past two decades, it will be necessary to intensify research with the aim of finding not only the precise epigenetic changes caused by the presence of diabetes in the gestational stage but also those mechanisms that cause them, with the aim of understanding how fetal programming occurs in association with the intrauterine environment [8,49,76].

Although an overview of the effects of diabetes on the epigenome during pregnancy has been provided, the limitations of this review should not be ignored. The studies carried out on this subject are numerous, and they could not all be included. Some reports do not specify whether a study was carried out with PGDM or GDM. Others do not address specific genes or epigenetic modifications. Yet others were not designed to determine gene expression. Finally, no longitudinal studies analyzing exposed offspring through different stages of life, from birth to adulthood, were found. Another important limitation is that it cannot be said whether the reported effects on the epigenome were caused solely by diabetes or whether other factors such as genetics, obesity, metabolic syndrome, or diet were also involved.

There is still much to study in the field of epigenetics. The knowledge obtained daily worldwide may be leading science to the development of therapies to prevent diabetes and other diseases of epigenetic origin, given that several studies have shown that epigenetic changes are potentially reversible. In this regard, ongoing research will be essential for a comprehensive understanding of the epigenome.

## 5. Conclusions

A strong connection has been observed between diabetes in pregnancy and alterations in the development of individuals exposed to it. Several authors have reported its association with an increased risk of developing abnormalities such as NTDs, expression changes in the regulation of specific genes, decreased insulin production and sensitivity, neurodevelopmental disorders, obesity, cardiac pathologies, and infertility, among others. These effects are associated with specific epigenetic changes in determined regions of the DNA and in certain histone proteins (e.g., H3K4 and H3K9), especially in conditions of hyperglycemia. 

As explained throughout this review, intrauterine development is a stage during which the individual is susceptible to environmental characteristics and changes that affect fetal programming. However, this stage should also be considered a window of opportunity for the diagnosis and possible prenatal treatment of the conditions that occur in diabetic pregnancies. Diabetes in pregnancy should not be underestimated but rather taken as a starting point for research that could help elucidate the origin of numerous ailments; this seems even more urgent with the prevalence of diabetes growing worldwide to become a critical health problem. This article presented a broad overview of the possible origins of diabetes during the gestational stage, determined based on multiple experimental models and studies in humans, for GDM and PGDM. Related maternal conditions, such as HFD, obesity, hyperglycemia, and IUGR, which are strongly associated with the development of diabetes in individuals, were discussed. Likewise, epigenetic modifications in DNA and histones that occur under the conditions mentioned above were reviewed to identify the effects of these changes in the decreased or increased expression of specific genes in tissues and organs.

The importance of epigenetics and its relationship to maternal conditions during pregnancy is a wide field of study that will lead to discoveries and scientific advances that will significantly improve the welfare of the global population. The continuous study of diabetes and its influence on the epigenome promises to help us understand the complex mechanisms with which it interferes in the pre-conception context and thus, in the future, develop methods to predict the susceptibility of an individual to various related diseases as well as techniques for their diagnosis and prevention.

## Figures and Tables

**Figure 1 biomedicines-12-00351-f001:**
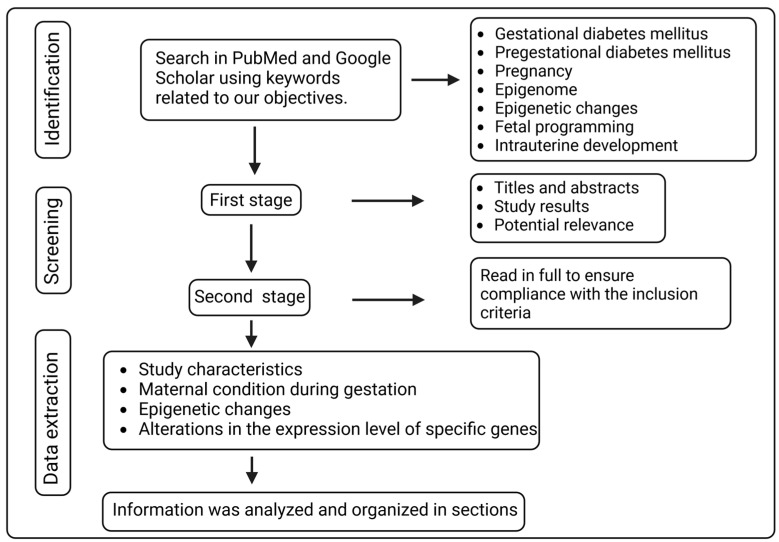
Flow diagram of the selection methodology for article inclusion.

**Figure 2 biomedicines-12-00351-f002:**
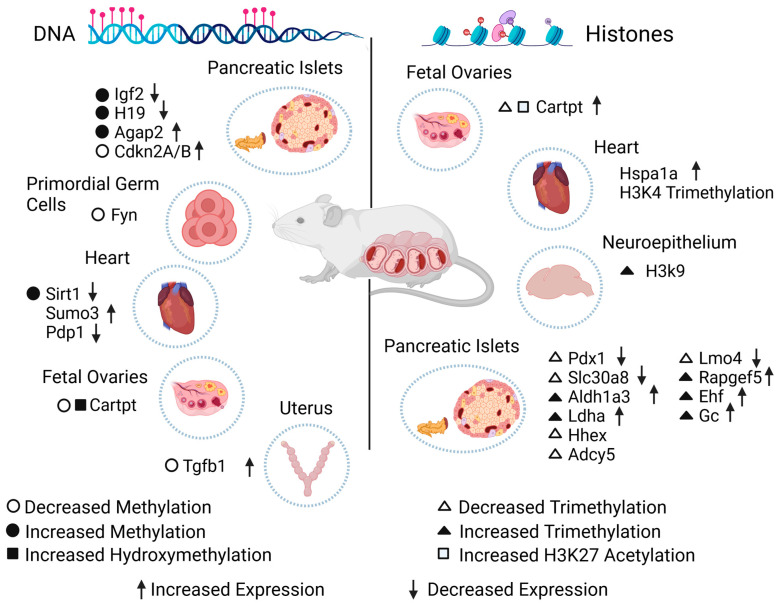
Effect of diabetes during gestation on the epigenome of exposed animal models. Epigenetic changes on DNA (**left**) or histones (**right**) in specific genes caused by diabetes during gestation. The scheme shows the type of epigenetic alteration found in cells, tissues, or organs, in addition to the change in the expression level of each gene. Igf2: insulin-like growth factor 2. H19: imprinted maternally expressed transcript. Agap2: ArfGAP with GTPase domain, ankyrin repeat, and PH domain 2. Cdkn2A/B: cyclin-dependent kinase inhibitor 2A/B. Fyn: Fyn proto-oncogene. Sirt1: sirtuin 1. Sumo3: small ubiquitin-like modifier 3. Pdp1: pyruvate dehydrogenase phosphatase catalytic subunit 1. Cartpt: cocaine- and amphetamine-regulated transcript Cart prepropeptide. Tgfb1: transforming growth factor beta 1. Hspa1a: heat shock protein 1A. Pdx1: pancreatic and duodenal homeobox. Slc30a8: solute carrier family 30 (zinc transporter), member 8. Aldh1a3: aldehyde dehydrogenase family 1, subfamily A3. Ldha: lactate dehydrogenase A. Hhex: hematopoietically expressed homeobox. Adcy5: adenylate cyclase 5. Lmo4: LIM domain only 4. Rapgef5: Rap guanine nucleotide exchange factor 5. Ehf: ETS homologous factor. Gc: vitamin D-binding protein.

**Figure 3 biomedicines-12-00351-f003:**
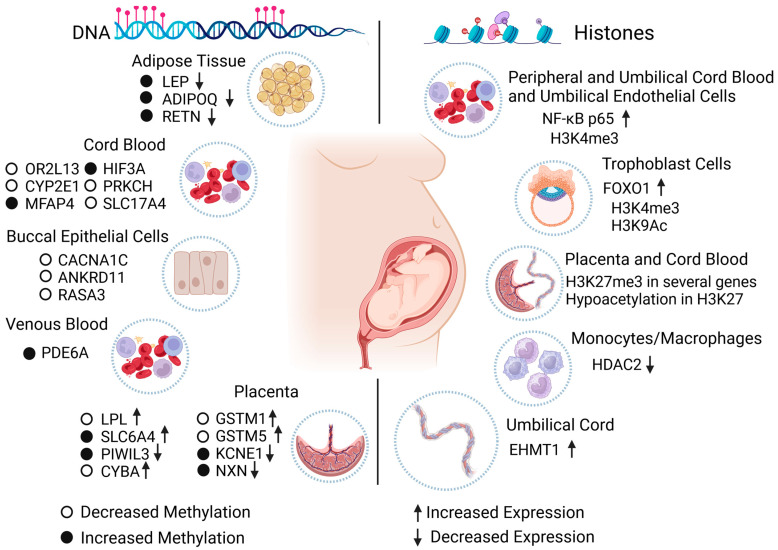
Effect of diabetes during gestation on the epigenome of individuals exposed in human studies. The scheme synthesizes the changes in the epigenome caused by the maternal condition of diabetes during pregnancy in cells, tissues, and human organs. Diabetes can cause epigenetic changes in DNA (**left**) and histones (**right**), which influence the proper expression of specific genes and potentially impact the health of exposed individuals in the short and long term. LEP: leptin. ADIPOQ: C1Q adiponectin and collagen domain containing. RETN: resistin. OR2L13: olfactory receptor family 2 subfamily L member 13. CYP2E1: cytochrome P450 family 2 subfamily E member 1. MFAP4: microfibril-associated protein 4. HIF3A: hypoxia-inducible factor 3 subunit alpha. PRKCH: protein kinase C eta. SLC17A4: solute carrier family 17 member 4. CACNA1C: calcium voltage-gated channel subunit alpha1 C. ANKRD11: ankyrin repeat domain-containing protein 11. RASA3: RAS p21 protein activator 3. PDE6A: phosphodiesterase 6A. LPL: lipoprotein lipase. SLC6A4: solute carrier family 6 member 4. PIWIL3: piwi-like RNA-mediated gene silencing 3. CYBA: cytochrome b-245 alpha chain. GSTM1: glutathione S-transferase mu 1. GSTM5: glutathione S-transferase mu 5. KCNE1: potassium voltage-gated channel subfamily E regulatory subunit 1. NXN: nucleoredoxin. NF-Kb p65: nuclear factor kappa-light-chain-enhancer of activated B cells p65. FOXO1: forkhead box O1. HDAC2: histone deacetylase 2. EHMT1: euchromatic histone-lysine methyltransferase 1.

**Table 1 biomedicines-12-00351-t001:** Epigenetic changes in DNA associated with maternal diet, GDM, or PGDM related to gene expression.

**Studies in Animals**					
**Experimental Model**	**Epigenetic Modification**	**Gene**	**Expression Effect**	**Tissue or Cell**	**Reference**
High-fat diet (HFD)	Hypomethylation and increased hydroxymethylation	LEP	Increased	Mouse plasma and fat pads	[26]
	Hypomethylation	Slc6a3	Increased	Mouse brains	[27]
		Oprm1			
		Penk			
	Hypermethylation	Pomc	Unaltered	Rats exposed to HFD before and after birth	[28]
	Reduced overall variation in DNA methylation	N/S	Changes in the expression of 3908 genes	Mouse liver tissue	[81]
GDM	Hypermethylation	Igf2	Downregulated	Mouse pancreatic islets and sperm	[23]
		H19			
	Decreased methylation	Fyn	N/S	Primordial germ cells from male mice	[75]
	Global DNA hypermethylation	Sirt1	Downregulated	Rat hearts	[76]
	Abnormal methylation	Sumo3	Increased	Genomic DNA of fetal heart tissue	[78]
		Pdp1	Decreased		
	Hypermethylation	Agap2	Upregulated	Mouse pancreas	[79]
	Hypomethylation	Cdkn2A/B promoter	Upregulated	Rat pancreatic islets	[80]
	Hypomethylation and increased hydroxymethylation	Cartpt promoter	No changes compared to controls	Mouse fetal ovaries	[82]
Undefined diabetes	Hypomethylation	Tgfb1	Increased	Rat uterus	[83]
PGDM	Hypermethylation	21 genes, see reference [73]	Decreased	Embryonic mouse hearts	[77]
**Studies in Humans**					
	**Epigenetic Modification**	**Gene**	**Expression Effect**	**Tissue or Cell**	**Reference**
Impaired glucose tolerance (IGT)	Hypermethylation	ADIPOQ	Decreased	Maternal side of placenta samples	[41]
	Hypermethylation in 20 fetal and 26 maternal CpG sites	LEP	Decreased	Placental tissue from fetal and maternal sides	[42]
Hyperglycemia	Hypomethylation and hypermethylation in YAP	SLIT1	Downregulated	Human fetal neural progenitor cells	[72]
		ROBO2			
		TAZ			
		YAP			
	Altered methylation in one or more CpG sites	RASD1	Increased	Human pancreatic islets	[74]
		GLRA1	Decreased		
		VAC14	Decreased		
		SLCO5A1	Increased		
		CHRNA5	Increased		
GDM	Hypomethylation	OR2L13 (Promoter)	N/S	Cord blood	[50]
		CYP2E1 (Gene body)			
	Hypermethylation	MFAP4	N/S	Fetal cord blood	[51]
	Hypermethylation	HIF3A			
	Hypomethylation	PRKCH			
	Hypomethylation	SLC17A4			
	Hypomethylation	LPL	Increased	Placenta samples	[58]
	Increased methylation	LEP and ADIPOQ	Decreased	Abdominal subcutaneous adipose tissue of adult offspring	[84]
	Hypermethylation	PDE6A (cg09109411)	N/S	Offspring venous blood	[85]
	Moderate to higher methylation	ADIPOQ	Decreased	Maternal subcutaneous and visceral adipose tissue and maternal and fetal blood	[86]
	Hypermethylation	SLC6A4	Increased	Fetal placenta side	[87]
Diabetes in pregnancy	Hypermethylation	PIWIL3	Decreased	Fetal placenta side	[88]
	Hypomethylation	CYBA	Increased		
	Hypomethylation	GSTM1	Increased		
	Hypomethylation	GSTM5	Increased		
	Hypermethylation	KCNE1	Decreased		
	Hypermethylation	NXN	Decreased		
	Global DNA hypomethylation	CACNA1C ANKRD11 RASA3	N/S	Offspring buccal epithelial cells	[89]
PGDM	Hypermethylation	RETN	Decreased	Abdominal subcutaneous adipose tissue of adult offspring	[84]

N/S: Not Specified.

**Table 2 biomedicines-12-00351-t002:** Epigenetic changes at the histone level associated with the maternal diet, GDM, or PGDM related to gene expression.

**Studies in Animals**					
**Experimental Model**	**Epigenetic Modification**	**Gene**	**Expression Effect**	**Tissue or Cell**	**Reference**
Intrauterine growth retardation (IUGR)	Fetal stage:	Pdx1		Rats with induced IUGR through the reduction of uteroplacental blood flow	[24]
	Deacetylation				
	Histone H3		Decreased		
	Histone H4		Decreased		
	After birth:				
	Demethylated H3K4		N/S		
	Methylated H3K9		N/S		
High-fat diet (HFD)	Decreased H3K4me3	Pfkl	Increased	Hepatoblasts and hepatocytes from mouse fetuses	[100]
		Pdha1			
		Ogdh			
		Acox1			
		Cpt1a			
	Increased H3K9 binding Oxtr at transcriptional start site	Oxtr	Increased	Mouse hippocampus	[102]
Hyperglycemia	H3K14ac	P300	Increased	Mouse embryos	[103]
	Deacetylation in H2K9	Sirt1	Decreased		
GDM	Decreased H3K27me3	Cartpt promoter	Increased	Mouse fetal ovaries	[82]
	Increased H3K27ac				
	H3K4me3	Hspa1a	Increased	Offspring rat hearts	[101]
PGDM	Decreased H3K4me3	Pdx1	Downregulated	Adult mouse pancreatic islets	[98]
	Decreased H3K4me3	Slc30a8	Downregulated		
	Increased H3K4me3	Aldh1a3	Upregulated		
	Increased H3K4me3	Ldha	Upregulated		
	Increased histone lysine malonylation in 21 histone lysine malonylation sites	N/S	N/S	Mouse brain cells	[99]
	Increased H3K9me3	N/S	N/S	Neuroepithelium from mice embryos	[104]
Multiparity-induced diabetes	Decreased H3K4me3	Hhex	N/S	Mice pancreatic islets with FOXO1 deficient gene	[97]
	Decreased H3K4me3	Adcy5	N/S		
	Decreased H3K4me3	Lmo4	Decreased		
	Increased H3K4me3	Rapgef5	Increased		
	Increased H3K4me3	Ehf	Increased		
	Increased H3K4me3	Gc	Increased		
**Studies in Humans**					
**Maternal Condition**	**Epigenetic Modification**	**Gene**	**Expression Effect**	**Tissue or Cell**	**Reference**
GDM	H3K4me3	NF-kB p65	Upregulated	Maternal peripheral blood mononuclear cells and umbilical vein endothelial and cord blood mononuclear cells from newborns	[95]
	H3K9ac	FOXO1	Downregulated	Trophoblast cells	[96]
	H3K4me3		No difference		
	H3K27me3	Several genes, see reference [97]	N/S	Cord blood and placenta from the fetal side	[105]
	N/S	HDAC2	Downregulated	Maternal peripheral blood monocytes/macrophages	[109]
Diabetes in pregnancy	Hypoacetylation H3K27 in males	N/S	N/S	Placental explants from the maternal side	[106]
PGDM	N/S	EHMT1	Upregulated	Umbilical cord	[108]

N/S: Not Specified.

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
