# Peer review of "Diabetes Mellitus and Pregnancy: An Insight into the Effects on the Epigenome"

_biomedicines, 2024, doi:10.3390/biomedicines12020351_

Round 1

Reviewer 1 Report (Previous Reviewer 1)

Comments and Suggestions for Authors

The manuscript has been improved and can be published.

Comments on the Quality of English Language

Moderate editing of English language required

Author Response

Reviewer 2 Report (New Reviewer)

Comments and Suggestions for Authors

Diabetes Mellitus and Pregnancy: An Insight at the Effects on the Epigenome is a good selection of
topic in current context
1. Abstract: You should give your insight about the impact you got from literature review
rather then telling we are going to discuss about those things.
2. Introduction: 1 st 4 para you have written about PGDM/GDM. Rather you should focus on
epigenetic changes due to GDM. How the science has evolved in this field till date.
In the last para you should mention your intended finding out of the this review and direction of
future research in this regard for the readers.
What are research going on to prevent this epigenetic modifications due GDM?
3. Methodology: Give a flow diagram for selection of article.
4. Give a result section after methodology
5. After results you should start discussion perspective
6. Limitation of your review need to be mentioned before conclusion
7. Is there any longitudinal study to support the epigenetic changes that occurred due to GDM
has manifested in the children?
8. What are other confounding factors that are responsible for epigenetic changes during
pregnancy sans GDM?
9. Cases of pregnancy with strict control of GDM, is there any epigenetic changes as
uncontrolled GDM?
10. Give a future perspective- how to prevent this harmful epigenetic changes in GDM.
11. Is there in relation with Hb A1c level in pregnancy with epigenetic changes?
12. Conclusion: should short and precise. It should not be descriptive.

Author Response

Reviewer 3 Report (New Reviewer)

Comments and Suggestions for Authors

The authors of the review article have done a great job. Because of the intricate text, It is recommended the authors seek the help of Moderate English Editing. Figures 1 and 2 are especially incomprehensible. The authors should explain the abbreviations below the figures.

Comments on the Quality of English Language

Because of the intricate text, It is recommended the authors seek the help of Moderate English Editing.

Round 2

Reviewer 2 Report (New Reviewer)

Comments and Suggestions for Authors

The authors revised the manuscript as desired. They have addressed the points raised, adequately. 

This manuscript is a resubmission of an earlier submission. The following is a list of the peer review reports and author responses from that submission.

Round 1

Reviewer 1 Report

Comments and Suggestions for Authors

The manuscript “Diabetes Mellitus and Pregnancy: An Insight at the Effects on the Epigenome” is a review article regarding the current state-of-art of epigenetic studies, the relationship with diabetes mellitus and the effects that occur between generations.

I appreciate the work performed by authors since the manuscript is generally well-written, provides a good view on the topic and can be of interest for the readers. However, there are important concerns that authors must address:

1.    In the context of diabetes and epigenetic modifications, it should be mentioned the possible role of enzymes such as nicotinamide N-methyltransferase (NNMT). NNMT has been demonstrated to be involved in the development of diabetes and diabetes-related disorders (PMID: 30552109; PMID: 34368359; PMID: 29688621).  NNMT can exert a great impact of epigenetics since it is a master regulator of intracellular NAD and SAM content (PMID: 36829935). Since NNMT can affect NAD homeostasis, NAD-dependent enzymes and concentration of SAM, it has a great impact on epigenetics, as demonstrated by Ulanovskaya et al. in an elegant study (PMID: 23455543). Moreover, the use of specific NNMT inhibitors has been suggested in the context of metabolic disorders (PMID: 33859793; PMID: 36104373; PMID: 34704059). All these considerations should be included and discussed in the manuscript.

2.    A figure regarding the impact of epigenetics on diabetes would enrich the manuscript.  

3.    Table 1 and table 2 are somehow difficult to be read. Perhaps, less information included in the table, or a more concise presentation would help the readers.

4.    The conclusion section should be expanded including author’s point of view.

Comments on the Quality of English Language

Moderate editing of English language required.

Reviewer 2 Report

Comments and Suggestions for Authors

The title of the paper is enticing, but what follows is an incomprehensible text both in form and content.  The text is also full of inappropriate borrowed sentences or half-sentences.  The literature search program is unclear.

- This seems to be a narrative review, but we are not informed on how the literature was assembled (search engine etc.).  How do we know the authors were not "cherry picking"?  We need a flow chart.

- Both the models and the tissues/cells are insufficiently explained, even the tables only mention "IUGR" "PGDM" "hyperglycemia" "IGT" "Diabetes in pregnancy".  Here and there, the authors mention "high-fat diet" or "streptozotocin", but these models are insufficiently explained in terms of how they work, what effects they have on the beta-cells and the fetus, etc.  For example: high-fat diets may cause similar or higher fetal weights, streptozotocin causes IUGR.  The targets in the tables go from cells and tissues (fine), but also include "rats exposed to HFD before and after birth" or "rats with induced IUGR through the reduction of the uteroplacental blood flow".

- The targets range from placenta, fetal/offspring brain, pancreas, heart and even gonads and uterus.  But we are uninformed on what morphological and/or functional changes maternal diabetes may cause in these tissues.  Here and there, "disease" or "autism" is suggested, but it all remains vague.  The age of the fetus or offspring might also be of importance.

- The English is extremely clunky, in the sense that it is next to incomprehensible.  I will not even begin to list all examples.  Worse, the language is sometimes stigmatizing: diabetes is apparently caused by "bad habits" (line 25), whereas T1DM (and T2DM to a large extent) have strong genetic (and indeed epi-genetic) roots.  DES is an inappropriate abbreviation for diabetic embryopathy syndrome, as it is commonly associated with the clinical DES syndrome (diethylstilbestrol). 

-" Cell models appear" to include in vitro and ex vivo models, but this is also true for the "clinical studies": cells or tissue A or B were prelevated.  Clinical "trials" are unavailable in this area, I think.  Thus, the whole structure of the paper needs to be re-organized.

- On page 5/20, the authors suddenly discuss MODY, which is a mono-GENETIC cause of diabetes par excellence.  Hence, we appear to be in a genetic-epigenetic soup.  It is also unclear what is meant by the mother-child binomial, maybe we should say a dyad.  Etc. 

- Citations are all right if a previous author has formulated something particularly well.  However, borrowed phrases and citations on every page of the text reek of plagiarism.

- The length and depth of the Discussion is not in line with the Results section.    

Comments on the Quality of English Language

Incomprehensible English: extremely clunky ++++
